# Design, Synthesis, and Molecular Docking Study of New Tyrosyl-DNA Phosphodiesterase 1 (TDP1) Inhibitors Combining Resin Acids and Adamantane Moieties

**DOI:** 10.3390/ph14050422

**Published:** 2021-05-01

**Authors:** Kseniya Kovaleva, Olga Yarovaya, Konstantin Ponomarev, Sergey Cheresiz, Amirhossein Azimirad, Irina Chernyshova, Alexandra Zakharenko, Vasily Konev, Tatiana Khlebnikova, Evgenii Mozhaytsev, Evgenii Suslov, Dmitry Nilov, Vytas Švedas, Andrey Pokrovsky, Olga Lavrik, Nariman Salakhutdinov

**Affiliations:** 1N. N. Vorozhtsov Novosibirsk Institute of Organic Chemistry, Siberian Branch of the Russian Academy of Sciences, Lavrentiev Ave. 9, 630090 Novosibirsk, Russia; kseniya.kovaleva3103@yandex.ru (K.K.); ponomarev@nioch.nsc.ru (K.P.); mozh@nioch.nsc.ru (E.M.); suslov@nioch.nsc.ru (E.S.); anvar@nioch.nsc.ru (N.S.); 2V. Zelman Institute for the Medicine and Psychology, Novosibirsk State University, Pirogova St. 1, 630090 Novosibirsk, Russia; cheresiz@yandex.ru (S.C.); azimirad@mail.ru (A.A.); chernyshova0305@gmail.com (I.C.); agpok@inbox.ru (A.P.); lavrik@niboch.nsc.ru (O.L.); 3Novosibirsk Institute of Chemical Biology and Fundamental Medicine, Siberian Branch of the Russian Academy of Sciences, Lavrentiev Ave. 8, 630090 Novosibirsk, Russia; a.zakharenko73@gmail.com; 4Boreskov Institute of Catalysis, Siberian Branch of the Russian Academy of Sciences, Lavrentiev Ave. 5, 630090 Novosibirsk, Russia; konvasnik@gmail.com (V.K.); khleb@catalysis.nsk.su (T.K.); 5Belozersky Institute of Physicochemical Biology, Lomonosov Moscow State University, Lenin Hills 1, Bldg. 40, 119991 Moscow, Russia; nilovdm@gmail.com (D.N.); vytas@belozersky.msu.ru (V.Š.); 6Faculty of Bioengineering and Bioinformatics, Lomonosov Moscow State University, Lenin Hills 1, Bldg. 73, 119991 Moscow, Russia

**Keywords:** tyrosil-DNA-phosphodiesterase 1, adamantane, resin acid, TDP1

## Abstract

In this paper, a series of novel abietyl and dehydroabietyl ureas, thioureas, amides, and thioamides bearing adamantane moieties were designed, synthesized, and evaluated for their inhibitory activities against tyrosil-DNA-phosphodiesterase 1 (TDP1). The synthesized compounds were able to inhibit TDP1 at micromolar concentrations (0.19–2.3 µM) and demonstrated low cytotoxicity in the T98G glioma cell line. The effect of the terpene fragment, the linker structure, and the adamantane residue on the biological properties of the new compounds was investigated. Based on molecular docking results, we suppose that adamantane derivatives of resin acids bind to the TDP1 covalent intermediate, forming a hydrogen bond with Ser463 and hydrophobic contacts with the Phe259 and Trp590 residues and the oligonucleotide fragment of the substrate.

## 1. Introduction

DNA in living organisms is constantly exposed to a variety of physical and chemical stresses, and damage occurs as a result. Bulk DNA damage is caused by UV light and environmental mutagens, and X-rays cause DNA double-strand breaks. Defects in the repair of DNA damage are implicated in a variety of diseases, many of which are typified by neurological dysfunction and/or increased genetic instability and cancer [1]. Traditional cancer chemotherapy is aimed at damaging the DNA of malignant cells, and the results depend on the effectiveness of their repair systems. Recently, compounds that act as DNA repair inhibitors have been considered as potential drugs [2,3]. The enzyme tyrosil-DNA-phosphodiesterase 1 (TDP1) is one of the promising ones [4]. This enzyme is an important supplementary target for anticancer therapies based on topoisomerase inhibitors 1 (TOP1), since it plays a key role in the removal of TOP1-DNA adducts stabilized by TOP1 inhibitors such as camptothecine [5] and its clinical derivatives [6]. TDP1 is also capable of hydrolysing apurinic sites, and thus leading to their repair. This may be the key activity needed for the repair of DNA damage caused by antitumour alkylating drugs such as temozolomide (TMZ), as well as ionising radiation [7]. Thus, the inhibition of TDP1 activity may significantly enhance the therapeutic effect of some anticancer agents. SCAN1 is a natural mutant of TDP1 where His493 is replaced with Arg493 in the binding pocket [8]. The mutation changes the geometry of the enzyme active site, and the enzyme remains covalently bound to DNA. This mutation leads to a severe neurodegenerative disease spinocerebellar ataxia syndrome with axonal neuropathy (SCAN1). It is currently suggested that the pathology is caused by the accumulation of the SCAN1-DNA covalent cleavage complexes [9]. It is assumed that nerve cells especially suffer from the accumulation of such adducts due to their nonproliferative nature leading to the progressive accumulation of unrepaired DNA lesions [10]. Therefore, suppression of SCAN1 activity could potentially improve the SCAN1 patients’ condition and prevent the progression of the disease. The search for inhibitors of key DNA repair enzymes is a promising area of medical chemistry, as it represents one of the ways to design effective therapies for cancer, as well as cardiovascular and neurodegenerative diseases. Recently, a number of TDP1 inhibitor structural classes have been studied, including pyrimidine nucleosides [11], furamidine [12], compounds with benzopentathiepine moiety [13], indenoisoquinolines [14], and 5-arylidenethioxothiazolidinones [15] (Figure 1).

Hybrid molecules created from different pharmacophores of natural and synthetic equivalents are successfully used in pharmaceutical practice [16]. New hybrid compounds have been synthesised starting from the pharmacophoric natural compounds with inhibitory properties against TDP1. These include phenolic usnic acid derivatives A [17], 7-hydroxycoumarins B [18], and 4-arylcoumarins C [19], derivatives of deoxycholic acid D [20] and adamantanecarboxylic acid monoterpene esters E [21] (Figure 2).

Our group previously obtained a set of ureas and thioureas based on the natural terpenoid dehydroabietylamine [22]. These compounds are able to inhibit TDP1 in the submicromolar range. They also lack toxicity against different cell lines in concentrations up to 100 μM. For the first time, we have shown that dehydroabietylamine TDP1 inhibitors in combination with TMZ demonstrate a better cytotoxic effect on glioblastoma cells than TMZ alone, taken at the same concentration. Compound **1** (Figure 3), which has a fragment of resin acid and adamantane, was an efficient inhibitor of TDP1 activity in vitro, and enhanced the cytotoxic effect of TMZ on glioblastoma cells. We synthesised a series of dehydroabietylamine derivatives containing the heterocyclic fragment 2-thioxoimidazolidin-4-ones and studied their activity against TDP1 [23]. It is important to note that not all of the synthesized heterocyclic derivatives are suitable for studying inhibitory activity, as some of the substances proved to be extremely insoluble. In addition, the combination of an adamantane fragment with terpenes of various structures has been shown to be successful in identifying new inhibitors of the TDP1 repair enzyme [23,24,25,26,27]. Since the combination of a terpene resin acid backbone with an adamantane fragment in compound **1** proved to be most successful and the compound had the most important biological properties, we set out in the present study to synthesise analogues of compound **1** to examine their structural activity. The design of the target derivatives is shown in Figure 3. It includes the variations of the linker type and length, diterpene, and adamantane moieties. Understanding which structural blocks are most important for the target biological activity and whether more active agents can be obtained by available synthetic methods is the main goal of the presented work.

## 2. Results and Discussion

### 2.1. Chemistry

Dehydroabietylamine (DHAm) is a diterpenic primary amine obtained from dehydroabietic acid (DHA). Dehydroabietic and abietic acids are components of resins of coniferous plants; for example, the high acid content is found in the resin of *Picea obovata* [27]. Dehydroabietylamine can be directly obtained from the resin by the reduction of dehydroabietyl nitrile.

The synthetic route for obtaining compounds **1**–**12** is shown in Scheme 1. A set of ureas and thioureas **1**–**4** was obtained by the interaction of dehydroabietylamine hydrochloride with 1- and 2- adamantyl isocyanates and isothiocyanates. The starting isocyanates and isothiocyanates were synthesized using the methods described earlier. In particular, 1-adamantyl isocyanate was obtained by the Curtius rearrangement of 1-adamantyl acyl azide formed in situ by interaction of corresponding acyl chloride with sodium azide [28]. Isomeric 2-adamantyl isocyanate was synthesized by the reaction of 2-adamantylamine hydrochloride with triphosgene in the presence of sodium hydrocarbonate, with dichloromethane used as a solvent [29]. Refluxing of 1-adamantaneamine with phenyl isothiocyanate in dry toluene resulted in 1-adamantane isothiocyanate [30]. To obtain 2-adamantyl isothiocyanate, 2-adamantaneamine hydrochloride was treated by triethylamine followed by carbon disulfide and DMAP/Boc_2_O subsequently [31]. For the present study, we re-synthesised compound **1**, which previously showed the best inhibitory characteristics and the ability to enhance the cytostatic properties of TMZ. Monosubstituted urea **5** was prepared by treating dehydroabietylamine hydrochloride with potassium cyanate. Amide and thioamide groups were considered as another variant of linker structurally similar to the ureas. The target amides **6** and **7** were obtained from dehydroabietylamine hydrochloride and 1- and 2-adamantanecarbonyl chlorides. The amide group of compound **6** was converted to thioamide using Lawesson’s reagent. The reaction proceeded under harsh conditions. Refluxing in toluene led to the formation of thioamide **8**. Compound **8** was isolated individually with a small yield. When the reaction was carried out in lower boiling solvents, the target product was not detected, even after a long period.

To obtain the target compounds **9**–**12**, starting with the dehydroabietic and abietic acids, the following synthetic route was taken. Following the three-step procedure described previously [32], nordehydroabietyl and norabietyl isocyanates were prepared. According to this method, treatment of the resin acids with SOCl_2_ afforded the corresponding chlorides, which were then converted to azides by interaction with NaN_3_. The azides underwent Curtius rearrangement by refluxing in toluene, resulting in decarboxylation and the formation of the corresponding isocyanates. Reaction of the obtained resin acid isocyanates with 1- and 2-adamantylamine hydrochlorides in the presence of a base provided good yields of the ureas **9**–**12** (80–90%). 

As a result of this work, compounds **2**–**12** (Scheme 1) were synthesised and characterized using physico-chemical methods. The ureas **1** and **2** (and thioureas **3** and **4**) differed from one another by the position of the adamantane fragment (1 and 2 respectively). Monosubstituted urea **5**, without the adamantane fragment in its structure, was prepared to clarify the contribution of this moiety to the studied compounds. Substances **6**–**8** have linkers of a different type. Ureas **9**–**10** differed from the leading compound **1** by lacking a CH_2_ group in the linker, while ureas **11**–**12**, in addition to the above, differed in the terpene part. 

### 2.2. TDP1 Assay and Cytotoxicity Studies

The primary screening of the inhibitory activities against TDP1 was performed using an in vitro cell-free system involving the recombinant TDP1 and a fluorescent reporter probe as previously described [13]. The IC_50_ values were found for derivatives **2**–**12** (and are presented in Table 1), a commercially available TDP inhibitor Furamidine was used as a reference drug [12]. We show here that the obtained compounds have the capacity to inhibit TDP1 in vitro within a 0.19–2.3 µM range. As can be seen from the data in the table, only compound **5**—monosubstituted urea (lacking the adamantane moiety)—showed no activity against TDP1. 

All the compounds with the exception of **5** were able to inhibit TDP1 at micromolar concentrations (0.19–2.3 µM). We studied the structure–activity relationship for a number of the compounds synthesized. After considering the effect of the diterpene fragment on the inhibitory characteristics, we conclude that ureas with dehydroabietyl **1**–**2** and nordehydroabietyl **9**–**10** backbone work in lower concentrations than with norabietyl **11**–**12**. For ureas **1**–**2** and **9**–**10**, the IC_50_ values were in the 0.19–0.8 µM range, and for ureas **11**–**12**, the IC_50_ values were higher—1.4–1.7 µM. However, ureas **9**–**12**, lacking a CH_2_ group in the terpene part, demonstrated extremely low solubility in water and almost all organic solvents, which does not make them promising for further study. The choice of 1-adamantane or 2-adamantane substituent did not significantly affect the inhibitory characteristics, but their absence negatively affected them. Monosubstituted urea **5** (without any bulky fragment) showed no activity at concentrations up to 15 µM. In a previous study [22], we showed that a decrease in the size of the substituent led to a decrease in activity, and the date obtain herein consistent with this. Among the compounds belonging to the urea, thiourea, amide, and thioamide classes, compound **8** with the thioamide linker inhibited TDP1 in the highest concentrations (IC_50_ = 2.3 µM). 

QSAR prediction methods offer a useful tool to identify drug-like compounds [33,34], and therefore we have calculated LogP values for synthesized inhibitors as main determinant of brain tissue binding. Octanol/water LogP predicted with GALAS algorithm [35] and QSAR software ACD/Percepta (www.acdlabs.com accessed on 29 April 2021) indicate that the obtained adamantane derivatives have similar lipophilicity (Table 1). The corresponding rate of brain penetration LogPS and extent of brain penetration LogBB, calculated using LogP, molecular size, and H-bonding parameters as inputs, are suitable for penetration into the central nervous system (see Appendix A).

Since dehydroabietylamine and its derivatives are known to possess high cytotoxicity against several cancer cells lines [36,37,38], the ureas, thioureas, amides, and thioamide (**1**–**12**) synthesised in the present study were tested against the T98G glioma cells. Since we used the T98G glioblastoma cell line for the first time for our experiments, we first attempted to perform the cytotoxicity study in a range of concentrations from 10 to 100 µM, as we did previously with the TDP1 inhibitory compounds when working with the other cell lines. However, the cytotoxicity at 50 and 100 µM of our compounds turned out to be rather high in the T98G cell line. We then measured the cytotoxicity of individual TDP1 inhibitors at 2.5, 5, 10, and 25 µM. The studied compounds were prepared as 50 mM stock solutions in DMSO and added to T98G glioma cells at 2.5 µM to 25 µM concentrations, either individually or in combination with 1000 or 2000 µM of TMZ. Preparation of stock solutions showed that the studied compounds varied in their solubility in DMSO. Compounds **9**, **10**, **11**, and **12** failed to dissolve either at 50 µM or at 10 µM concentrations. They were therefore discarded from the later cytotoxicity studies.

When individual compounds were added to T98G glioma cells at 2.5, 5, 10, and 25 µM concentrations, they demonstrated moderate toxicity. The cell viabilities at 2.5 and 5 µM lay within the 90–100% range (Figure 4).

To investigate the cytotoxicity of combinations of the studied compounds with TMZ, we combined 5 µM concentrations of each with 1000 µM or 2000 µM of TMZ and compared their toxicity with 1000 µM and 2000 µM of TMZ alone. At 1000 µM, TMZ was almost non-toxic to our cell culture, with ~95% of cells surviving the treatment. Adding 5 µM of the compounds to 1000 µM of TMZ did not increase the cytotoxicity to the glioma cells. Higher cytotoxicity was obtained with 2000 µM of TMZ, which inhibited cell viability by ~40%. Combining the compounds with 2000 µM of TMZ resulted in considerably higher toxicity (i.e., an increase of 5–15%) compared with TMZ alone, which indicated the additive profile of the action of TDP1 inhibitors with TMZ. The effects of combinations of dehydroabietylamine derivatives **1**–**4** and **6**–**8** with TMZ on T98G viability are shown in Figure 5. 

### 2.3. Molecular Docking Studies

The reaction catalyzed by TDP1 proceeds in two steps: the nucleophile His263 residue attacks the 3′-phosphotyrosyl bond of the substrate, and the His493 residue activates the water molecule to cleave the covalent intermediate [39,40,41]. This offers two potential therapeutic strategies: (1) the inhibition of the first step to prevent the formation of the 3′-phosphohistidine intermediate; and (2) the inhibition of the second step to prevent the intermediate hydrolysis [4,42]. Using molecular docking, we tested the discussed resin acid derivatives against both the molecular model of the apo form and that of the covalent intermediate. The inhibitors were found to bind preferentially to the intermediate structure, as demonstrated in Figure 6.

A resin acid fragment occupies part of the peptide binding site—peptide is released upon the intermediate formation—and forms hydrophobic contacts with the side chain of Trp590. An adamantane fragment interacts with both the Phe259 residue of the oligonucleotide binding site and with methylene (ribose) and the methyl (nucleobase) groups of the oligonucleotide. A carbamide linker forms a hydrogen bond with the Ser463 side chain, whilst its NH groups are orientated towards the solvent.

We conclude that adamantane derivatives of resin acids stabilize the TDP1 intermediate (covalent complex of TDP1 with DNA) in a manner that is analogous with the stabilisation of topoisomerase–DNA covalent complexes by camptothecins [43,44]. It is worth noting that compound **5**, which lacked the adamantane fragment, failed to inhibit TDP1. Using our proposed model, this can be explained as follows: monosubstituted urea has an additional hydrogen bond donor, the –NH_2_ group, which is orientated towards a hydrophobic adamantane-binding region. This may result in the unfavourable interaction of compound **5** with the TDP1 intermediate.

## 3. Materials and Methods

### 3.1. Chemistry

All reagents and solvents were purchased from commercial sources and were used as received without further purification. Reactions were monitored by thin-layer chromatography (TLC) in silica gel. The TLC plates were visualised by exposure to ultraviolet light (254 and 365 nm). Merck (Merck KGaA, Darmstadt, Germany) silica gel (63–200 μm) was used for column chromatography. The ^1^H and ^13^C NMR spectra in CDCl_3_, CD_3_OD, and DMSO-*d*_6_ were recorded on a Bruker AV-400 spectrometer (400.13 and 100.61 MHz, respectively, Bruker, Billerica, MA, USA). The residual signals of the solvent were used as references (*δ*H 7.24, *δ*C 76.90 for CDCl_3_; *δ*H 2.50, *δ*C 39.50 for DMSO-d6). High-resolution mass spectra were recorded on a Thermo Scientific DFS instrument (Thermo Fisher Scientific Inc., Waltham, MA, USA) in full scan mode over the *m/z* range of 0–500 by ionisation with an electron impact of 70 eV, and direct introduction of samples. IR spectra were recorded on a Vector22 spectrometer (KBr, Bruker, Billerica, MA, USA). Thin-layer chromatography was performed on Silufol plates (UV-254, Merck KGaA, Darmstadt, Germany). The atomic numbering in the compounds is provided for the assignment of signals in the NMR spectra and is different from the atomic numbering in the systematic name. The analytical and spectroscopic studies were conducted at the Chemical Service Center for the collective use of the Siberian Branch of the Russian Academy of Sciences (SB RAS).

#### 3.1.1. General Procedure for the Synthesis of Ureas and Thioureas **1**–**4**

Dehydroabietylamine hydrochloride (0.5 g, 1.55 mmol) and triethylamine (0.28 mL, 2.0 mmol) were dissolved in CHCl_3_ (25 mL) and an equimolar amount of the appropriate adamantyl isocyanate or isothiocyanate was added. The reaction mixture was stirred on a magnetic stirrer for 24 h at room temperature. Conversion was monitored by TLC. The reaction mixture was washed with 10 mL of distilled water. The organic layer was dried over anhydrous Na_2_SO_4_ and filtered. The resultant liquid was evaporated under vacuum. The residue was purified using column chromatography on silica gel with CHCl_3_ as an eluent and a MeOH gradient from 0 to 100%. 

***N*-abieta-8,11,13-trien-18-yl-*N*’-1-adamantylurea** (**1**). The spectral data for the compound **1** has been described previously [22].

***N*-abieta-8,11,13-trien-18-yl-*N*’-2-adamantylurea** (**2**). Yield 72%, white powder. M.p. 150 °C. IR (KBr) νmax 3361, 2908, 1629, 1562 cm^−1^. ^1^H NMR (400MHz, CDCl_3_, δ, ppm, J/Hz): 6.86 (1H, s, H–14), 6.96 (1H, d, *J*_11, 12_ = 8.2, H–12), 7.14 (1H, d, *J*_11, 12_ = 8.2, H–11), 5.06 and 5.26 (1H both, s, NH), 0.87 (3H, s, Me–19), 1.20 (6H, d, *J*_16, 15_ = 6.9, Me–16 and Me–17), 1.18 (3H, s, Me–20), 2.80 (1H, sept, *J*_15, 16_ = 6.9, H–15), 2.82–2.95 (2H, m, 2H–7), 2.24 (1H, d, ^2^*J* = 12.3, H–1*e*), 3.72–3.83 (1H, m, H–23), 2.95–3.13 (2H, m, H–18), 1.65–1.91 (15H, m, H–22, H–26, H-27, H-28, 2H-25, 2H-24, 2H-30, 2H-29, H-3*e*, H-3*a*, H-6*e*), 1.27-1.65 (7H, m, 2H-31, H-5*a*, H-6*a*, H-2*a*, H-2*e*, H–1*a*). ^13^C NMR (100MHz, CDCl_3_, δ, ppm): 158.10 (C-20), 147.21 (C-9), 145.31 (C-13), 134.77 (C-8), 126.68 (C-14), 124.05 (C-11), 123.59 (C-12), 18.43 (Me-19), 23.85 (Me-17 and Me-16), 25.16 (Me-20), 27.19 and 27.04 (C-22, C-26), 32.48 and 32.43 (C-27, C-28), 33.28 (C-15), 45.03 (C-5), 53.92 (C-23), 18.58 (C-2), 18.70 (C-6), 30.05 (C-7), 31.60 and 31.57 (C-25, C-30), 50.61 (C-18), 37.30 and 37.27 (C-23, C-28), 38.29 (C-4), 37.51 (C-3), 37.16 (C-10), 37.06 (C-1), 38.36 (C–30). Found, *m*/*z*: 462.3613 [M]^+^. C_31_H_46_ON_2_. Calculated, *m*/*z*: 462.3605. 

***N*-abieta-8,11,13-trien-18-yl-*N*’-1-adamantylthiourea** (**3**). Yield 70%, white powder. M.p. 103 ^o^C. IR (KBr) νmax 3265, 2908, 1538 cm^–1 1^H NMR (400MHz, CDCl_3_, δ, ppm, J/Hz): 6.88 (1H, s, H–14), 6.95 (1H, d, *J*_11, 12_ = 8.1, H–12), 7.13 (1H, d, *J*_11, 12_ = 8.1, H–11), 0.98 (3H, s, Me–19), 1.18 (6H, d, *J*_16, 15_ =6.9, Me–16 and Me–17), 1.20 (3H, s, Me–20), 2.79 (1H, sept, *J*_15, 16_ = 6.9, H–15), 2.82–2.98 (2H, m, 2H–7), 2.29 (1H, d, ^2^*J* = 13.0, H–1*e*), 2.00–2.21 (4H, m, H–26, H–27, H–28, H–6*a*), 1.83–1.99 (8H, m, 2H-23, 2H-24, 2H-25, H-6*e*, H-2*a*), 3.23-3.40 and 3.65-3.84 (2H, m, H-18), 1.36–1.44 (1H, d, ^2^*J* = 13.0, H-1*a*), 1.45-1.81 (9H, m, 2H-29, 2H-30, 2H-31, H-3, H-2*e*, H–5*a*). ^13^C NMR (100MHz, CDCl_3_ + CD_3_OD, δ, ppm): 181.35 (C-21), 146.72 (C-9), 145.42 (C-13), 134.17 (C-8), 126.48 (C-14), 123.75 (C–11), 123.51 (C-12), 18.26 (Me-19), 23.60 and 23.69 (Me-17 and Me-16), 24.66 (Me-20), 33.15 (C-15), 46.00 (C-5), 29.04 (C-26, C–27, C-28), 18.29 (C-2), 18.72 (C-6), 29.66 (C-7), 38.21 (C-4), 37.28 (C-1), 37.14 (C–10), 36.54 (C-3), 53.66 (C-18), 35.66 (C-29, C-30, C-31), 42.00 (C-23, C-24, C-25), 56.15 (C–22). Found, *m*/*z*: 478.3380 [M]^+^. C_31_H_46_N_2_S. Calculated, *m*/*z*: 478.3376.

***N*-abieta-8,11,13-trien-18-yl-*N*’-2-adamantylthiourea** (**4**). Yield 75%, white powder. M.p. 116 ^o^C. IR (KBr) νmax 3278, 2908, 1537 cm^−1^. ^1^H NMR (400MHz, CDCl_3_, δ, ppm, J/Hz): 6.86 (1H, s, H–14), 6.95 (1H, d, *J*_11, 12_ = 8.1, H–12), 7.12 (1H, d, *J*_11, 12_ = 8.1, H–11), 6.94 and 5.21 (1H both, s, NH), 0.95 (3H, s, Me–18), 1.19 (6H, d, *J*_16,15_ = 6.9, Me–16 and Me–17), 1.19 (3H, s, Me–19), 2.79 (1H, sept, *J*_15,16_ =6.9, H–15), 2.92–2.92 (2H, m, 2H–7), 4.05 (1H, br s, H–23), 3.09-3.62 (2H, m, 2H-18), 2.26 (1H, ^2^*J* = 12.6, H–1*e*), 1.94-2.10 (2H, m, H-22, H-26), 1.26–1.51 (4H, m, H-6, 2H-2, H–1*a*), 1.51-1.93 (16H, m, 2H-24, 2H-29, 2H-25, 2H-31, 2H-30, H–27, H-28, H-6, 2H-3, H–5). ^13^C NMR (100MHz, CDCl_3_, δ, ppm): 181.06 (C-21), 146.72 (C-9), 145.47 (C-13), 134.37 (C-8), 126.67 (C-14), 123.89 (C-11), 123.65 (C-12), 18.44 (Me-19), 23.87 and 23.82 (Me-17 and Me-16), 25.02 (Me-20), 33.24 (C-15), 45.76 (C-5), 26.78 (C-27, C-28), 31.58 (C-22, C-26), 57.83 (C-23), 55.31 (C-18), 18.40 (C-2), 18.93 (C-6), 29.86 (C-7), 31.72 (C-25, C-31), 38.11 (C-4), 37.57 (C-1), 36.57 (C-30), 36.78 (C-24, C-29), 37.26 and 37.16 (C-10 and C–3). Found, *m*/*z*: 478.3368 [M]^+^. C_31_H_46_N_2_S. Calculated, *m*/*z*: 478.3376. 

#### 3.1.2. Synthesis of Urea **5**


Dehydroabietylamine hydrochloride (0.34 g, 1.06 mmol) was dissolved in EtOH (30 mL), and an aqueous solution of potassium cyanate (0.1 g of KNCO in 5 mL of water) was added. The mixture was refluxed for 6 h, then cooled to room temperature. The solvent was evaporated under vacuum. The solid residue was dissolved in CHCl_3_ (20 mL) and washed with water (10 mL) and 5% aqueous NaOH solution (10 mL). The urea was purified using column chromatography on silica gel with CHCl_3_ as an eluent and a MeOH gradient from 0 to 20%.

***N*-abieta-8,11,13-trien-18-ylurea** (**5**). Yield 46%, white powder. M.p. 108 ^o^C. IR (KBr) νmax 3430, 2927, 1652 cm^−1^. ^1^H NMR (400MHz, CDCl_3_, δ, ppm, J/Hz): 6.86 (1H, s, H–14), 6.96 (1H, d, *J*_11, 12_ = 8.2, H–12), 7.14 (1H, d, *J*_11, 12_ = 8.2, H–11), 0.89 (3H, s, Me–19), 1.19 (6H, d, *J*_16, 15_ = 6.9, Me–16 and Me–17), 1.18 (3H, s, Me–20), 2.25 (1H, d, ^2^*J* = 12.3, H–1*e*), 2.79 (1H, sept, *J*_15, 16_ = 6.9, H–15), 2.93–3.00 and 3.02–3.11 (1H both, m, H–18), 2.80–2.95 (2H, m, H–7), 1.77–1.94 (2H, m, H–6e, H–3a), 1.52–1.76 (3H, m, H–2*e*, H–3*e*, H–1*a*), 1.26–1.50 (3H, m, H–6*e*, H–2*a* H–5*a*), 4.53 (2H, s, NH_2_), 4.96 (1H, s, NH). ^13^C NMR (100MHz, CDCl_3_, δ, ppm): 159.18 (C–21), 147.17 (C–9), 145.47 (C–13), 134.72 (C–8), 126.72 (C–14), 124.02 (C–11), 123.65 (C–12), 23.85 (Me–17 and Me–16), 25.05 (Me–20), 33.29 (C–15), 44.87 (C–5), 29.95 (C–7), 38.28 (C–4), 37.27 (C–1, C–10), 35.89 (C–3), 50.73 (C-18), 18.76 (Me–19, C–2, C–6). Found, *m*/*z*: 328.2503 [M]^+^. C_21_H_32_ON_2_. Calculated, *m*/*z*: 328.2509. 

**Norabietyl isocyanate.** Yield 56%, light-yellow oil. IR (KBr) νmax 2933, 2250, 1459 cm^−1^. ^1^H NMR (400MHz, CDCl_3_, δ, ppm, J/Hz): 5.77 (1H, s, H-14), 5.39-5.44 (1H, m, H-7), 1.00 and 0.99 (3H both, д, J_16, 15_=6.9, Me-16 and Me-17), 0.75 (3H, s, Me-18), 1.34 (3H, s, Me-19), 2.21 (1H, sept, J_15, 16_=6.9, H-15), 2.27 (1H, d, J=18.2, H-5), 1.08 (1H, dt, J=3.8, J=13.3, H-11a), 0.81-0.88 (1H, m, H-1a), 1.15-1.29 (2H, m, H-2a, H-2e), 1.42-1.48 (1H, m, H-3), 1.54-1.70 (3H, m, H-1e, H-11e, H-9), 1.74-1.80 (1H, m, H-3e), 1.80-1.86 (1H, m, H-6a), 1.86-2.03 (3H, m, H-6e, H-12a, H-12e). ^13^C NMR (100MHz, CDCl_3_, δ, ppm): 145.18 (C-13), 135.22 (C-8), 122.13 (C-14), 119.97 (C-7), 21.17 and 20.62 (Me-17 and Me-16), 23.66 (Me-18), 13.32 (Me-19), 34.66 (C-15), 50.60 (C-9), 51.30 (C-5), 19.18 (C-2), 27.20 (C-12), 22.54 (C-11), 23.86 (C-6), 35.59 (C-10), 38.00 (C-1), 43.14 (C-3), 61.24 (C-4), 121.92 (C-20). Found, *m*/*z*: 299.2240 [M]^+^. C_20_H_29_ON. Calculated, *m*/*z*: 299.2244. 

#### 3.1.3. General Procedure for the Synthesis of Amides **6**–**7**


Dehydroabietylamine hydrochloride (1.0 g, 3.1 mmol) was mixed with an equimolar amount of 1- or 2-adamantanecarbonyl chloride (0.62 g, 3.1 mmol) in 30 mL of CH_3_CN with the addition of Et_3_N (0.56 mL, 4.0 mmol). The reaction mixture was stirred on a magnetic stirrer for 24 h at room temperature. Upon completion, the solvent was evaporated under vacuum. The solid residue was dissolved in CHCl_3_ (20 mL) and washed with water (15 mL). The organic layer was dried over anhydrous Na_2_SO_4_ and filtered. The resultant liquid was evaporated under vacuum. The residue was purified using column chromatography on silica gel with hexane/ethyl acetate system, with a concentration gradient (EtOAc 0–25%) as an eluent.

***N*-abieta-8,11,13-trien-18-yladamantan-1-carboxamide** (**6**). Yield 50%, white powder. M.p. 90 ^o^C. IR (KBr) νmax 3363, 2906, 1639, 1525 cm^−1^. ^1^H NMR (400MHz, CDCl_3_, δ, ppm, J/Hz): 6.88 (1H, d, *J*_12,14_=1.7, H-14), 6.98 (1H, dd, *J*_11, 12_=8.2, *J*_12,14_=1.7, H-12), 7.16 (1H, d, *J*_11, 12_=8.2, H-11), 0.91 (3H, s, Me-19), 1.21 (6H, d, *J*_16, 15_=6.9, Me-16 and Me-17), 1.20 (3H, s, Me-20), 2.81 (1H, sept, *J*_15, 16_=6.9, H-15), 2.28 (1H, d, ^2^*J*=12.3, H-1*e*), 3.17-3.22 and 3.08-3.13 (1H both, m, H-18), 2.85-2.91 and 2.73-2.79 (1H both, m, H-7), 1.99-2.02 (3H, m, H-26, H-27, H-28), 1.80-1.83 (6H, m, H-23, H-24, H-25), 1.84-1.89 (1H, m, H-6e), 1.42 (1H, d, ^2^*J*=13.0, H-3*e*), 1.32-1.39 (2H, m, H-5*a*, H-1*a*), 1.62-1.75 (8H, m, H-29, H-30, H-31, H-3a, H-2*e*), 1.22-1.30 (2H, m, H-6*e*, H-2*a*). ^13^C NMR (100MHz, CDCl_3_, δ, ppm): 177.7 (C-21), 146.9 (C-9), 145.4 (C-13), 134.6 (C-8), 126.8 (C-14), 124.1 (C-11), 123.7 (C-12), 18.4 (Me-19), 23.84 and 23.80 (Me-17 and Me-16), 25.4 (Me-20), 33.3 (C-15), 46.2 (C-5), 28.0 (C-26, C-27, C-28), 18.5 (C-2), 18.9 (C-6), 30.4 (C-7), 36.4 (C-29, C-30, C-31), 39.3 (C-23, C-24, C-25), 38.3 (C-4), 37.5 (C-1), 37.3 (C-10), 36.2 (C-3), 49.5 (C-18), 40.8 (C-22). Found, *m*/*z*: 447.3490 [M]^+^. C_31_H_45_ON. Calculated, *m*/*z*: 447.3496. 

***N*-abieta-8,11,13-trien-18-yladamantan-2-carboxamide** (**7**). Yield 46%, white powder. M.p. 94 ^o^C. IR (KBr) νmax 3311, 2904, 1642, 1542 cm^−1^. ^1^H NMR (400MHz, CDCl_3_, δ, ppm, J/Hz):6.87 (1H, s, H-14), 6.97 (1H, d, *J*_11, 12_=8.1, H-12), 7.15 (1H, d, *J*_11, 12_=8.1, H-11), 0.92 (3H, s, Me-19), 1.20 (6H, d, *J*_16, 15_=6.9, Me-16 and Me-17), 1.19 (3H, s, Me-20), 2.80 (1H, sept, *J*_15, 16_=6.9, H-15), 2.27 (1H, d, ^2^*J*=12.7, H-1*e*), 2.73-2.94 (2H, m, H-7), 5.59 (1H, s, NH), 3.13-3.26 (2H, m, H-18), 2.39-2.47 (1H, m, H-23), 2.16-2.23 (2H, m, H-22, H-26), 1.63-1.80 (7H, m, H-24, H-29, H-25, H-31, H-30, H-3, H-6), 1.52-1.63 (3H, m, H-5*a*, H-1e, H-3), 1.80-2.00 (6H, m, H-24, H-29, H-25, H-31, H-28, H-27), 1.29-1.46 (4H, m, H-6, 2H-2, H-1*a*). ^13^C NMR (100MHz, CDCl_3_, δ, ppm): 173.9 (C-21), 147.0 (C-9), 145.5 (C-13), 134.7 (C-8), 126.8 (C-14), 124.0 (C-11), 123.7 (C-12), 18.5 (Me-19), 23.8 and 23.9 (Me-17 and Me-16), 25.2 (Me-20), 33.3 (C-15), 45.6 (C-5), 50.0 (C-23), 29.96 and 30.04 (C-22, C-26), 27.24 and 27.36 (C-27, C-28), 18.5 (C-2), 18.9 (C-6), 30.2 (C-7), 49.5 (C-18), 33.16 and 33.21 (C-25, C-31), 38.20 and 38.26 (C-25, C-31, C-4), 36.3 (C-3), 37.22, 37.31, 37.33 (C-10, C-1, C-30). Found, *m*/*z*: 447.3503 [M]^+^. C_31_H_45_ON. Calculated, *m*/*z*: 447.3500. 

#### 3.1.4. Synthesis of Thioamide **8**


Amide **6** (0.4 g, 0.9 mmol) and Lawesson’s reagent (0.18 g, 0.45 mmol) were refluxed in *o*-xylene (20 mL) for 3 h. Conversion was monitored by TLC. The solvent was removed under vacuum. The residue was purified using column chromatography on silica gel with CHCl_3_ as an eluent and a MeOH gradient from 0 to 20%.

***N*-abieta-8,11,13-trien-18-yladamantan-1-carbothioamide (8).** Yield 10%, light-yellow powder. M.p. 166 ^o^C. IR (KBr) νmax 3386, 2904, 1525 cm^−1^. ^1^H NMR (400MHz, CDCl_3_, δ, ppm, J/Hz): 6.89 (1H, s, H-14), 6.99 (1H, d, *J*_11, 12_=8.2, H-12), 7.15 (1H, d, *J*_11, 12_=8.2, H-11), 7.42 (1H, s, NH), 0.99 (3H, s, Me-19), 1.21 (6H, d, *J*_16, 15_=6.9, Me-16 and Me-17), 1.22 (3H, s, Me-20), 2.81 (1H, sept, *J*_15, 16_=6.9, H-15), 2.72-2.93 (2H, m, 2H-7), 2.31 (1H, m, H-1*e*), 3.75-3.85 (1H, m, H-18), 3.48-3.56 (1H, m, H-18), 2.04-2.14 (3H, m, H-26, H-27, H-28), 1.50-1.56 (1H, m, H-3*a*), 1.91-2.03 (7H, m, 2H-23, 2H-24, 2H-25, H-6*e*), 1.28-1.47 (3H, m, H-5*a*, H-1*a*, H-3*e*), 1.60-1.91 (9H, m, 2H-28, 2H-29, 2H-30, H-6*a*, H-2*a*, H-2*e*). ^13^C NMR (100MHz, CDCl_3_, δ, ppm): 212.88 (C-21), 146.27 (C-9), 145.28 (C-13), 134.11 (C-8), 126.56 (C-14), 123.84 (C-11), 123.56 (C-12), 18.33 (Me-19), 23.54 and 23.49 (Me-17 and Me-16), 25.08 (Me-20), 32.97 (C-15), 46.75 (C-5), 28.14 (C-26, C-27, C-28), 18.18 (C-2), 18.79 (C-6), 30.04 (C-7), 35.93 (C-29, C-30, C-31), 41.53 (C-23, C-24, C-25), 37.85 (C-4), 37.27 (C-1), 37.18 (C-10), 36.56 (C-3), 56.12 (C-18), 46.08 (C-22). Found, *m*/*z*: 463.3264 [M]^+^. C_31_H_45_NS. Calculated, *m*/*z*: 463.3267.

#### 3.1.5. General Procedure for the Synthesis of Norabietyl and Nordehydroabietyl Ureas 

Norabietyl or nordehydroabietyl isocyanate (0.3 g, 1.0 mmol) was dissolved in CHCl_3_ (15 mL). An equimolar amount (0.19 g, 1.0 mmol) of 1- or 2-adamantylamine hydrochloride with triethylamine (0.17 mL, 1.2 mmol) was dissolved in EtOH (15 mL) and added to isocyanate solution. The reaction mixture was stirred on a magnetic stirrer for 24 h at room temperature. The precipitated norabietyl ureas were filtered off and were not additionally purified. The nordehydroabietyl urea solutions were washed with water (15 mL) and dried over Na_2_SO_4_. The solvent was removed in vacuo. The solid residues were recrystallized from acetonitrile.

***N*-1-adamantyl-*N*’-[(1*R*,4a*S*,10a*R*)-7-isopropyl-1,4a-dimethyl-1,2,3,4,4a,9,10,10a- octahydrophenanthren-1-yl]urea** (**9**). Yield 85%, white powder. M.p. 235 ^o^C. IR (KBr) νmax 3357, 2906, 1629, 1554 cm^−1^. ^1^H NMR (400MHz, CDCl_3_ + CD_3_OD, δ, ppm, J/Hz): 6.76 (1H, s, H–14), 6.87 (1H, d, *J*_11, 12_ = 8.2, H–12), 7.05 (1H, d, *J*_11, 12_ = 8.2, H–11), 1.07 (3H, s, Me–18), 1.11 (6H, d, *J*_16, 15_ = 6.9, Me–16 and Me–17), 1.10 (3H, s, Me–19), 2.71 (1H, sept, *J*_15, 16_ = 6.9, H–15), 2.74-2.83 (2H, m, 2H–7), 2.10 (2H, m, H–1*e*, H–5*a*), 1.88–2.01 (4H, m, H–25, H–26, H–27, H–6*e*), 1.73–1.86 (8H, m, 2H–22, 2H–23, 2H–24, H–3*e*, H–3*a*), 1.47–1.66 (9H, m, 2H–28, 2H–29, 2H–30, H–6*a*, H–2*a*, H–2*e*), 1.30-1.40 (1H, m, H–1*a*). ^13^C NMR (100MHz, CDCl_3_ + CD_3_OD, δ, ppm):157.17 (C–20), 146.74 (C–9), 145.16 (C–13), 134.44 (C-8), 126.44 (C–14), 124.08 (C–11), 123.49 (C–12), 20.76 (Me–18), 23.62 and 23.66 (Me–17 and Me–16), 24.77 (Me–19), 33.15 (C–15), 46.93 (C–5), 29.25 (C–25, C-26, C-27), 18.59 (C–2), 19.46 (C–6), 30.11 (C–7), 36.18 (C–28, C-29, C-30), 42.13 (C–22, C-23, C-24), 55.77 (C–4), 37.89 (C–1), 37.50 and 37.46 (C–10 and C-3), 50.04 (C-21). Found, *m*/*z*: 448.3445 [M]^+^. C_30_H_44_ON_2_. Calculated, *m*/*z*: 448.3448. 

***N*-2-adamantyl-*N*’-[(1*R*,4a*S*,10a*R*)-7-isopropyl-1,4a-dimethyl-1,2,3,4,4a,9,10,10a-octahydrophenanthren-1-yl]urea** (**10**). Yield 80%, white powder. M.p. 233 ^o^C. IR (KBr) νmax 3357, 2912, 1623, 1556 cm^−1^. ^1^H NMR (400MHz, CDCl_3_, δ, ppm, J/Hz): 6.78 (1H, s, H–14), 6.89 (1H, d, *J*_11, 12_ = 8.2, H–12), 7.07 (1H, d, *J*_11, 12_ = 8.2, H–11), 1.20 (3H, s, Me–18), 1.11 (6H, d, *J_16_*_, 15_ = 6.9, Me–16 and Me–17), 1.10 (3H, s, Me–19), 2.73 (1H, sept, *J*_15, 16_ = 6.9, H–15), 2.74–2.87 (2H, m, 2H–7), 2.15 (1H, d, ^2^*J* = 12.3, H–1*e*), 3.64 (1H, s, H–22), 1.29–1.41 (1H, m, H–1*a*), 1.42–1.54 (2H, m, H–24, H–29), 1.94–2.08 (2H, m, H–23, H–28), 1.78–1.93 (2H, m, H–24, H–29), 1.55–1.65 (4H, m, H–26, H–27, 2H–30), 1.65–1.78 (11H, m, 2H–2, 2H–3, 2H–6, H–5, H–21, H–25, H–23, H–28). ^13^C NMR (100MHz, DMSO-*d*_6_, δ, ppm, J/Hz): 157.25 (C–20), 147.74 (C–9), 145.54 (C–13), 134.99 (C-8), 127.00 (C–14), 124.69 (C–11), 124.16 (C–12), 21.91 (Me–18), 24.14 and 24.48 (Me–17 and Me–16), 25.25 (Me–19), 46.53 (C–5), 30.43 (C–7), 18.89 (C–2), 20.06 (C–6), 55.76 (C–4), 53.25 (C-22), 27.46 and 27.52 (C-21, C-25), 31.80 and 31.92 (C-24, C-29), 32.86 and 32.94 (C-26, C-27), 33.43 (C–15), 37.57 (C-3), 37.73 (C–10), 38.32 and 38.36 (C–1, C-30), 37.97 and 37.92 (C-23, C-28). Found, *m*/*z*: 448.3449 [M]^+^. C_30_H_44_ON_2_. Calculated, *m*/*z*: 448.3448. 

***N*-1-adamantyl-*N*’-[(1*R*,4a*R*,10a*R*)-7-isopropyl-1,4a-dimethyl-1,2,3,4,4a,4b,5,6,10,10a-decahydrophenanthren-1-yl]urea** (**11**). Yield 90%, white powder. M.p. 224 ^o^C. IR (KBr) νmax 3346, 2906, 1633, 1560 cm^−1^. ^1^H NMR (400MHz, CDCl_3_ + CD_3_OD, δ, ppm, J/Hz): 5.65 (1H, s, H–14), 5.30 (1H, s, H–7), 0.88 and 0.89 (3H both, d, *J*_16, 15_=6.9, Me–16 and Me–17), 0.67 (3H, s, Me–18), 1.12 (3H, s, Me–19), 2.10 (1H, sept, *J*_15, 16_ = 6.9, H–15), 0.95–1.10 (2H, m, H–1a, H–11a), 1.32–1.48 (2H, m, H–2a, H–2e), 1.63–1.73 (2H, m, H–3a, H–11e), 1.48–1.57 (6H, m, 2H–29, 2H–30, 2H–28), 1.72–1.87 (9H, m, 2H–22, 2H–23, 2H–24, H–1e, H–3e, H–9), 1.87–2.05 (8H, m, H-25, H-26, H–27, H–5, H–6a, H–6e, 2H–12). ^13^C NMR (100MHz, CDCl_3_ + CD_3_OD, δ, ppm): 156.99 (C–20), 145.02 (C–13), 135.33 (C-8), 122.19 (C–14), 120.64 (C–7), 29.32 (C–25, C-26, C-27), 36.22 (C–28, C-29, C-30), 42.21 (C–22, C–23, C–24), 21.10 and 20.53 (Me–17 and Me–16), 21.38 (Me–18), 13.61 (Me–19), 34.65 (C–15), 50.65 (C–9), 47.00 (C–5), 19.15 (C–2), 27.20 (C–12), 35.44 (C–10), 38.14 (C–1), 37.92 (C–3), 22.49 (C–11), 23.50 (C–6), 55.45 (C–4), 50.17 (C–21). Found, *m*/*z*: 450.3600 [M]^+^. C_30_H_46_ON_2_. Calculated, *m*/*z*: 450.3605. 

***N*-2-adamantyl-*N*’-[(1*R*,4a*R*,10a*R*)-7-isopropyl-1,4a-dimethyl-1,2,3,4,4a,4b,5,6,10,10a-decahydrophenanthren-1-yl]urea** (**12**). Yield 80%, white powder. M.p. 213 ^o^C. IR (KBr) νmax 3395, 2908, 1629, 1556 cm^−1^. ^1^H NMR (400MHz, CDCl_3_ + CD_3_OD, δ, ppm, J/Hz): 5.62 (1H, s, H–14), 5.27 (1H, s, H–7), 0.88 and 0.87 (3H both, d, *J*_16, 15_ = 6.9, Me–16 and Me–17), 0.67 (3H, s, Me–18), 1.13 (3H, s, Me–19), 2.08 (1H, sept, *J*_15, 16_ = 6.9, H–15), 0.93–1.10 (2H, m, H–1a, H–11a), 1.32–1.48 (4H, m, H–2a, H–2e, H–24, H–29), 1.54–1.61 (2H, m, 2H–30), 1.89–2.05 (5H, m, H–5, H–6a, H–6e, 2H–12), 1.73–1.89 (3H, m, H–1e, H–3e, H–9), 1.60–1.75 (10H, m, H–3a, H–11e, H–26, H–27, H–21, H–25, 2H–23, 2H–28, H–24, H–29), 3.59 (1H, s, H–22). ^13^C NMR (100MHz, CDCl_3_ + CD_3_OD, δ, ppm): 157.29 (C–20), 144.82 (C–13), 135.25 (C–8), 122.11 (C–14), 120.52 (C–7), 13.53 (Me–19), 21.03 and 20.47 (Me–17 and Me–16), 21.28 (Me–18), 22.42 (C–11), 23.40 (C–6), 27.13 (C–12), 27.05 and 26.90 (C–21, C-25), 31.37 (C–24, C-29), 32.38 and 32.40 (C–26, C–27), 34.55 (C–15), 35.35 (C–10), 37.05 and 37.03 (C–23, C–28), 38.10 (C–1), 37.92 (C–3), 37.35 (C–30), 46.87 (C–5), 50.68 (C–9), 55.33 (C–4), 53.10 (C–22). Found, *m*/*z*: 450.3604 [M]^+^. C_30_H_46_ON_2_. Calculated, *m*/*z*: 450.3605.

### 3.2. TDP1 Assay 

The recombinant TDP1 was purified to homogeneity by chromatography on Ni-chelating resin and phosphocellulose P11 as previously described [45], using plasmid pET 16B-TDP1, kindly provided by Dr. K.W. Caldecott (University of Sussex, United Kingdom). The TDP1 activity measurements were carried out as described [13]. Briefly, TDP1-biosensor with fluorofore (FAM) at the 5′-end and a fluorescence quencher (BHQ1) at the 3′-end at a final concentration of 50 nM were incubated in a 200 μL volume that contained TDP1 buffer (50 mM Tris-HCl pH 8.0, 50 mM NaCl, and 7 mM β-mercaptoethanol) supplemented with purified 1.5 nM TDP1 and various concentrations of inhibitor. Fluorescence measurements (Ex485/Em520 nm) were carried out during the linear phase of the reaction (from 0 to 8 min for TDP1) every 55 sec. The reactions were incubated at a constant temperature of 26 °C in a POLARstar OPTIMA fluorimeter (BMG LABTECH, GmbH). The influence of compounds was evaluated by comparing the fluorescence increase rate in the presence of compounds with that of DMSO control wells. The data were imported into the MARS Data Analysis 2.0 program (BMG LABTECH), and the IC_50_ values (the concentration of a compound required to reduce the enzyme activity by 50%) were calculated. The TDP1-biosensor 5′-(5,6 FAM-aac gtc agg gtc ttc c-BHQ1)-3′ was synthesised in the Laboratory of Biomedical Chemistry, Institute of Chemical Biology and Fundamental Medicine, Novosibirsk, Russia. 

### 3.3. Cytotoxicity Experiments

Individual TDP1 inhibitors were prepared as 50 mM stock solutions in DMSO and were added to the cells at 2.5, 5, 10, or 25 μM. Temozolomide was prepared as 200 mM stock solution in DMSO and was added at 1 mM or 2 mM concentrations, either alone or in combination with TDP1 inhibitors. T98G and SNB19 glioma cells were maintained in DMem/F12 medium supplemented with 10% foetal bovine serum, l-glutamine, and penicillin/streptomycin and were split at 10,000 cells/well into the 96-well plates for cytotoxicity experiments. The drugs or drug combinations were incubated with cells for 72 h, then the MTT reagent (3-(4,5-dimethylthiazol-2-yl)-2,5-diphenyltetrazolium bromide) was added for 4 h. The produced purple formazan dye reporting the activity of cellular oxireductases was dissolved overnight in 10% solution of acidified SDS and the absorbance was measured using a Tecan plate reader. Each concentration of individual substance or each drug combination was tested in triplicate. The data obtained were processed using MS Excel software and presented as histogram plots.

### 3.4. Molecular Docking

The models of the apo form and covalent intermediate of human TDP1 were based on the 1NOP crystal structure [46] and constructed as reported in our previous work [47,48]. Molecular docking of inhibitors was performed with Lead Finder 1.1.15 [49,50]. An energy grid box with edges of 35 Å was centred on the N^ε2^ atom of the catalytic residue His263 and overlapped the active site and adjacent cavities. In our previous study, the TDP1 substrate-binding groove was identified and mapped based on the 1NOP structure (covalent complex with substrate analogue), and successfully tested in docking runs with diazaadamantane derivatives as reference TDP1 inhibitors [48]. Docking was done using a genetic algorithm in ‘extra precision’ mode. The protein structure was rigid, whereas rotating functional/rotatable groups of ligands was allowed. VMD 1.9.2 was used to visualise molecular structures [51].

## 4. Conclusions

As a result of this study, we synthesized a set of compounds containing the diterpene fragment. The effect of the terpene structural blocks, the length and structure of the ureide linker, and the site of attachment of the adamantane residue on the biological properties of the new abietylamine-based compounds were investigated, and in particular their ability both to inhibit the DNA repair enzyme TDP1 and to enhance the cytotoxic effect of TMZ. In this library of compounds, we studied the structure of compounds with demonstrated biological activity. The choice of 1-adamantane or 2-adamantane substituent did not significantly affect the inhibitory characteristics, but their absence negatively affected them. Ureas on nordehydroabietyl and norabietyl isocyanates lacking a CH_2_ group in the terpene part, demonstrated extremely low solubility in water and almost all organic solvents, which does not make them promising for further study. The starting compound **1** was the most effective in the inhibition of TDP1 and compound **2** was the most effective in the sensitization of glioma T98G cells to TMZ. We found some synergistic effects on cells T98G when using repair enzyme inhibitors, but they are not as high as we expected. At the same time, the substances we described could be of considerable interest when studied on other cancer cell lines or simultaneously with other cytostatics.

## Data Availability

Not applicable.

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
