# Peer review of "Design, Synthesis, and Molecular Docking Study of New Tyrosyl-DNA Phosphodiesterase 1 (TDP1) Inhibitors Combining Resin Acids and Adamantane Moieties"

_pharmaceuticals, 2021, doi:10.3390/ph14050422_

Round 1

Reviewer 1 Report

The article entitled: “Design, synthesis, and molecular docking study of new tyrosyl-DNA phosphodiesterase 1 (TDP1) inhibitors combining resin acids and adamantane moieties” concerns the development of potential inhibitors of TDP1, which could support the anticancer therapy through synergistic effect with other anticancer  drugs. In my opinion, the obtained compounds showed too weak biological activity, and the scope of biological research is too narrow to be published in the Pharmaceuticals journal. In a journal such as Pharmaceuticals, the developed molecules should exhibit an exceptional/outstanding biological activity that distinguish them from other less active compounds. The authors of the publication did not show that the compounds obtained by them show better properties than those reported before. I suggest to transfer this article to a more general journal like Molecules.
1. General comments:
1.1. Enzymatic activity
The authors previously reported the synthesis and biological properties of compound 1 (reference 11). The authors claimed, that compound 1, used as a reference compound in the present study, inhibited the TDP1 with IC50 = 0.19 μM. All new compounds 2-13, showed significantly weaker inhibition effect in the range of 0.59 (compound 9) to 2.3
μM (compound 8). No improvement was observed with respect to the previously published results in enzymatic activity.
2.2. Cytotoxic effect
The authors determined the cytotoxic effect of the obtained compounds for only one cell line. Such study show neither the potency nor the selectivity of the compounds tested and is meaningless for research. To determine the cytotoxic profile of the tested compounds (obligatory for new anticancer molecules), a panel containing cell lines
derived from tumor and normal cell lines should be used. The use of cell lines derived from cancer cells leads to the determination of the potency of the compounds, and the use of cell lines derived from normal cells allows to determine the selectivity of substances.
The authors must test their compounds on a panel of cell lines to precisely determine the cytotoxic effect of the newly synthesized molecules. IC50 values should be calculated and presented for the tested compounds and cell
lines. It facilitate the comparison of cytotoxic effect of compound tested.
2.3. Synergistic effect
Initially, the authors tested the synergistic effect using 1 mM of Temozolomid (TMZ, alkylating agent) and 5 μM of synthesized compounds, but didn’t observe significant effect on tested cell line. Increasing the concentration of TMZ to 2 mM resulted in the appearance of a stronger cytotoxic effect of the tested system, but it was still quite weak (5-15%), taking into account the high concentration of TMZ. All the newly obtained compounds showed a similar effect, no clear differentiation was observed 2 between the tested compounds. This weak synergistic effect may be partly due to the fact that the compounds obtained are weak TDP1 inhibitors, as mentioned earlier. In the introduction section of their article (line 37-40), the authors described that TDP1 inhibitors may have a beneficial synergistic effect with topoisomerase inhibitors such as camptothecine. Since the biological effect of TDP1 inhibitors, important in the
treatment of cancer, is largely dependent on and related to the therapeutic effect of other anti-cancer drugs, the authors should check the synergistic effect of the newly obtained compounds with camptothecine (what the authors suggested in the introduction).
2.4. Concusion
The conclusion section is very short. The first sentence is a summary, not a conclusion. This section lack of any important information. Please rewrite and expand this section, think about more conclusive statements.
2. Minor comments:
line 43-47
…The search for inhibitors of key DNA repair enzymes is a promising area of medical chemistry, as it represents one of the ways to design effective therapies for cancer, as well as cardiovascular and neurodegenerative diseases.
The authors describe only the potential use of inhibitors of key DNA repair enzymes in cancer treatment. They did not describe their potential use in cardiovascular and neurodegenerative diseases. A proper description of this topic, commentary and literature references should be added to the introduction.
line 89: dichloromethane
line 99-103:
The authors tried to introduce a thioamide group using Lawesson reagent, but got rather poor results. In the future research, I suggest to try some other thiolation reagents such as Davy-reagent p-tolyl (CAS: 114234-09-2) and its analogues. In my experiments (synthesis of tiolactams from lactams) I got better results with Davy-reagents then in the
case of Lawesson reagent.
line 172-176:
… Compounds 7 and 8 formed visible micelles when stock solutions were dissolved in a cell growth medium, as shown by light microscopy. The cytotoxicity of these compounds was studied; however, the concentrations of their solutions may be significantly different from those indicated in dilutions. In my opinion, if compound is not soluble in the culture medium, precipitates from the solution, and the real concentration of compound tested is not known, its meaningless to present the cytotoxic effect. This effect is always related to the concentration of the compound tested. If the concentration of the tested compound is unknown, the effect is not reliable.
3
line 215-217:
… We conclude that adamantane derivatives of resin acids stabilise the TDP1 intermediate in a manner that is analogous with the stabilisation of topoisomerase−DNA covalent complexes by camptothecins[31,32].
This fragment is not fully understandable to me. It seems that the authors compare the stabilizing effect of the ligand on TDP1 (protein) with the stabilizing effect of camptothecins on covalent topoisomerase-DNA adducts. Please, clarify.
Experimental part: add melting points for the newly obtained compounds, when applicable
Experimental part: mL, not ml
Line 321: please check molecular formula, one N is missing?
Line 531: 1 … 2 (bold)
Line 529-530: Bioorg. Med. Chem.
Line 546: Bioorg. Med. Chem.
Line 564: Russ. J. Bioorg. Chem.
Line 570: Anti-Cancer Agents Med. Chem.
Line 585: Bioorg. Med. Chem.
Ref. 19: add patent number. Cite the patent according to the publisher's
recommendations.
Add a space before any references throughout the text of your article.

Author Response

The article entitled: “Design, synthesis, and molecular docking study of new tyrosyl-DNA phosphodiesterase 1 (TDP1) inhibitors combining resin acids and adamantane moieties” concerns the development of potential inhibitors of TDP1, which could support the anticancer therapy through synergistic effect with other anticancer drugs. In my opinion, the obtained compounds showed too weak biological activity, and the scope of biological research is too narrow to be published in the Pharmaceuticals journal. In a journal such as Pharmaceuticals, the developed molecules should exhibit an exceptional/outstanding biological activity that distinguish them from other less active compounds. The authors of the publication did not show that the compounds obtained by them show better properties than those reported before. I suggest to transfer this article to a more general journal like Molecules.

We appreciate the thoughtful comments provided by the reviewer. These comments have helped us to significantly improve our manuscript.

Apparently, we have not described the purpose of this paper accurately enough. At the moment, we have a leader compound (agent 1) that is highly active against the repair enzyme. We have shown that when this compound is administered orally simultaneously with temozolovide, there is a significant reduction in the rate of tumor growth in animal models. These are not yet published results, just being prepared for publication. However, from the point of view of medicinal chemistry, it is necessary to understand whether we have exactly the leading compound in the class of substances under study. Understanding which structural blocks are most important for the target biological activity and whether more active agents can be obtained by available synthetic methods is the main goal of the presented work.

We fully agree with the reviewer, indeed, the compounds described by us have not shown better activity than agent 1. However, we believe that the information presented in this work may be important for both medical chemists and molecular biologists, specialists in the field of work with repair enzymes.

We have added a more detailed description of the purpose of the study in the introduction.

  1. General comments:

 1.1. Enzymatic activity

            The authors previously reported the synthesis and biological properties of compound 1 (reference 11). The authors claimed, that compound 1, used as a reference compound in the present study, inhibited the TDP1 with IC50 = 0.19 μM. All new compounds 2-13, showed significantly weaker inhibition effect in the range of 0.59 (compound 9) to 2.3 μM (compound 8). No improvement was observed with respect to the previously published results in enzymatic activity.

We agree with the remark, but we would like to point out that the activity of the agents we synthesized is higher than that of furamidine, commercial Tdp1 inhibitor. We have added the activity data of the indicated comparison drug to the table 1

 2.2. Cytotoxic effect

The authors determined the cytotoxic effect of the obtained compounds for only one cell line. Such study show neither the potency nor the selectivity of the compounds tested and is meaningless for research. To determine the cytotoxic profile of the tested compounds (obligatory for new anticancer molecules), a panel containing cell lines derived from tumor and normal cell lines should be used. The use of cell lines derived from cancer cells leads to the determination of the potency of the compounds, and the use of cell lines derived from normal cells allows to determine the selectivity of substances. The authors must test their compounds on a panel of cell lines to precisely determine the cytotoxic effect of the newly synthesized molecules.

Following the request of the Reviewer, we added another glioblastoma cell line, SNB19 (a derivative of the U373 cell line) to our cytotoxicity study. Unfortunately, the immortalized normal human astrocyte cell lines or the primary astrocyte culture sources are not available to our lab, thus, it was not possible to meet the other request of the reviewer to test the cytotoxicity in normal astrocytes.

IC50 values should be calculated and presented for the tested compounds and cell lines. It facilitate the comparison of cytotoxic effect of compound tested.

Since the 50% inhibition of cell metabolism (IC50) has been achieved for only one of our substances in either of the two cell lines (T98G and SNB19), the comparison of IC50 values for the studied panel of substances was not feasible.

 2.3. Synergistic effect

Initially, the authors tested the synergistic effect using 1 mM of Temozolomid (TMZ, alkylating agent) and 5 μM of synthesized compounds, but didn’t observe significant effect on tested cell line. Increasing the concentration of TMZ to 2 mM resulted in the appearance of a stronger cytotoxic effect of the tested system, but it was still quite weak (5-15%), taking into account the high concentration of TMZ. All the newly obtained compounds showed a similar effect, no clear differentiation was observed 2 between the tested compounds. This weak synergistic effect may be partly due to the fact that the compounds obtained are weak TDP1 inhibitors, as mentioned earlier.

We agree that the concentrations of TMZ, we used, are rather high, but, in our experience, these glioblastoma cell lines are actually quite resistant to TMZ treatment. 1 mcM is almost non-toxic to them, and to achieve 2 mcM, we had to make stock solutions of TMZ more concentrated, than proposed by the manufacturer (200mcM vs 125 mcM), in order not to overload cells with DMSO. Meanwhile, even that high TMZ concentration does not reach IC50.

We also find that 5-15% improvements in efficiency are not high, but the effect of TMZ itself, as a concurrent co-therapy to standard radiotherapy, is only 20% (increased median overall survival from 12.1 to 14.6 months), as well //R. Stupp, W. P. Mason, M. J. van den Bent et al., “Radiotherapy plus concomitant and adjuvant temozolomide for glioblastoma,” The New England Journal of Medicine, vol. 352, no. 10, pp. 987–996, 2005.]. Taking this consideration into account, we find even such moderate improvements to the efficiency of the current anti-GBM drugs, worth reporting.

In the introduction section of their article (line 37-40), the authors described that TDP1 inhibitors may have a beneficial synergistic effect with topoisomerase inhibitors such as camptothecine. Since the biological effect of TDP1 inhibitors, important in the treatment of cancer, is largely dependent on and related to the therapeutic effect of other anti-cancer drugs, the authors should check the synergistic effect of the newly obtained compounds with camptothecine (what the authors suggested in the introduction).

However, since topoisomerase inhibitors are not approved as the first- or second-line chemotherapy against malignant glioma, we did not study their combinations with our substances in glioblastoma cell lines. We have previously performed experiments with topotecan on other cancer cell lines, but this is not the subject of this paper and we do not think it is appropriate to add these results to the manuscript.

 2.4. Concusion

The conclusion section is very short. The first sentence is a summary, not a conclusion. This section lack of any important information. Please rewrite and expand this section, think about more conclusive statements.

We have considerably expanded the conclusion of the paper, adding more detail on the main results.

  1. Minor comments:

line 43-47

…The search for inhibitors of key DNA repair enzymes is a promising area of medical chemistry, as it represents one of the ways to design effective therapies for cancer, as well as cardiovascular and neurodegenerative diseases. The authors describe only the potential use of inhibitors of key DNA repair enzymes in cancer treatment. They did not describe their potential use in cardiovascular and neurodegenerative diseases. A proper description of this topic, commentary and literature references should be added to the introduction.

We have added to the introduction information about SCAN1 - a mutant form of TDP1, as well as neurodegenerative pathologies caused by this mutation. Suppression of SCAN1 activity could potentially improve the SCAN1 patients’ condition and prevent the progression of the disease.

line 89: dichloromethane

We have corrected this mistake

line 99-103: The authors tried to introduce a thioamide group using Lawesson reagent, but got rather poor results. In the future research, I suggest to try some other thiolation reagents such as Davy-reagent p-tolyl (CAS: 114234-09-2) and its analogues. In my experiments (synthesis of tiolactams from lactams) I got better results with Davy-reagents then in the case of Lawesson reagent

We thank the reviewer for a valuable suggestion. We will try to use this reagent in our next research.

line 172-176:

… Compounds 7 and 8 formed visible micelles when stock solutions were dissolved in a cell growth medium, as shown by light microscopy. The cytotoxicity of these compounds was studied; however, the concentrations of their solutions may be significantly different from those indicated in dilutions. In my opinion, if compound is not soluble in the culture medium, precipitates from the solution, and the real concentration of compound tested is not known, its meaningless to present the cytotoxic effect. This effect is always related to the concentration of the compound tested. If the concentration of the tested compound is unknown, the effect is not reliable.

We have removed this phrase from the main text. At the same time, we would like to point out that the problem of solubility of organic compounds in biological tests is extremely important and is far from always easy to solve. Often researchers do not describe in any detail the solubility of the compounds tested, whereas we have honestly described the effect we have seen.

3 line 215-217: … We conclude that adamantane derivatives of resin acids stabilise the TDP1 intermediate in a manner that is analogous with the stabilisation of topoisomerase−DNA covalent complexes by camptothecins[31,32]. This fragment is not fully understandable to me. It seems that the authors compare the stabilizing effect of the ligand on TDP1 (protein) with the stabilizing effect of camptothecins on covalent topoisomerase-DNA adducts. Please, clarify.

 Thank you for this comment. In this paragraph we compare the possible stabilizing effect of our resin acid derivatives on covalent TDP1-DNA adduct with the effect of camptothecins on covalent topoisomerase-DNA adducts. Via docking we have found that resin acids preferentially bind to the TDP1 intermediate, a covalent complex of TDP1 with DNA, and conclude that it can somehow resemble the manner of action of camptothecins against the covalent complex of topoisomerase with DNA. To clarify, the corresponding comment has been added in parentheses: “We conclude that adamantane derivatives of resin acids stabilise the TDP1 intermediate (covalent complex of TDP1 with DNA) in a manner that is analogous with the stabilisation of topoisomerase−DNA covalent complexes by camptothecins”.

Experimental part: add melting points for the newly obtained compounds, when applicable

Experimental part: mL, not ml

 Line 321: please check molecular formula, one N is missing?

Line 531: 1 … 2 (bold)

 Line 529-530: Bioorg. Med. Chem.

 Line 546: Bioorg. Med. Chem.

 Line 564: Russ. J. Bioorg. Chem.

 Line 570: Anti-Cancer Agents Med.

Chem. Line 585: Bioorg. Med. Chem.

Ref. 19: add patent number. Cite the patent according to the publisher's recommendations.

Add a space before any references throughout the text of your article.

We thank the reviewer for carefully reading our manuscript. We added melting points for new compounds 2-12, replaced ml with mL throughout the text, corrected the molecular formula of the urea 5, corrected the reference list according to the journal's recommendations, and added spaces before the references.

Reviewer 2 Report

The manuscript is for the synthesis of resin acids and adamantane hybrids, evaluation of their TDP1 inhibitory activities including molecular docking, and combination of selected hybrids with the antitumor alkylation agent, temozolomide, for the treatment of T98G glioma cells. First of all, the design and synthesis of hybrid chemicals are presented well in a view of scientific soundness. All the synthesized chemicals except for compound 5 which lacks adamantane moiety showed potent TDP1 inhibitory activities. Unfortunately, their potent TDP1 inhibitory activities did not guarantee significant cytotoxicities of combination therapy with temozolomide against T98G glioma cells, but a slight enhancement of has been achieved as shown in Figure 3. Therefore, I recommend publication of this manuscript in Pharmaceuticals after appropriate revisions. Suggested revision: In Scheme 1, reaction temperatures are missing. Add proper number for each “t”.

Author Response

The manuscript is for the synthesis of resin acids and adamantane hybrids, evaluation of their TDP1 inhibitory activities including molecular docking, and combination of selected hybrids with the antitumor alkylation agent, temozolomide, for the treatment of T98G glioma cells. First of all, the design and synthesis of hybrid chemicals are presented well in a view of scientific soundness. All the synthesized chemicals except for compound 5 which lacks adamantane moiety showed potent TDP1 inhibitory activities. Unfortunately, their potent TDP1 inhibitory activities did not guarantee significant cytotoxicities of combination therapy with temozolomide against T98G glioma cells, but a slight enhancement of has been achieved as shown in Figure 3. Therefore, I recommend publication of this manuscript in Pharmaceuticals after appropriate revisions. Suggested revision: In Scheme 1, reaction temperatures are missing. Add proper number for each “t”.

We are grateful to the reviewer for the comment provided. In accordance with the recommendation, we added the reaction temperatures to the Scheme 1.

Reviewer 3 Report

The manuscript "Design, synthesis, and molecular docking study of new tyrosyl- 2 DNA phosphodiesterase 1 (TDP1) inhibitors combining resin acids and adamantane moieties" seems to be interesting even if it is sometimes poorly discussed and detailed. In my opinion, revisions are necessary prior to pubblication.

Abstract- please modify it in order to be more informative. Revise the last sentence including the main information they achieved through their dockign studies.

Introduction- please the sentences "Recently, compounds that act as DNA 
repair inhibitors have been considered as potential drugs[1,2]. The enzyme tyrosil-DNA- 36 phosphodiesterase 1 (TDP1) is one of the promising ones[3]". should be accompanied by a scheme about the main chemical scaffold so far developed in this context.

All the following sentence "Hybrid molecules created from different pharmacophores of natural and synthetic 48
equivalents are successfully used in pharmaceutical practice[7]. New hybrid compounds 49
have been synthesised by combining the pharmacophoric moiety of a set of natural com- 50
pounds with inhibitory properties against TDP1. These include phenolic usnic acid and a set of different monoterpenoid fragments[8], 7-hydroxycoumarin and monoterpenoid moieties[9], and 4-arylcoumarin and monoterpenoid fragments[10]. Our group previously obtained a set of ureas and thioureas based on the natural terpenoid dehydroabietylamine[11]". should be supported by the proper chemical series. Please, the related figures.

Figure 1- it seems to be unclear as well as the rationale of this study.

Please, the introduction should be concluded with the aims of this study.

Results and discussion- 2.2. TDP1 assay and cytotoxicity studies

Please, remove the PK and toxicity prediction by ACD/Percepta and move it in a specific paragraph. Please discuss more in details this kind of in silico study. Refer to the following papers:

New Insights into the Binding Features of F508del CFTR Potentiators: A Molecular Docking, Pharmacophore Mapping and QSAR Analysis Approach.

Righetti G, Casale M, Tonelli M, Liessi N, Fossa P, Pedemonte N, Millo E, Cichero E.Pharmaceuticals (Basel). 2020 Dec 4;13(12):445. doi: 10.3390/ph13120445   Synthesis and Biological Evaluation of Novel (thio)semicarbazone-Based Benzimidazoles as Antiviral Agents against Human Respiratory Viruses. Francesconi V, Cichero E, Schenone S, Naesens L, Tonelli M.Molecules. 2020 Mar 25;25(7):1487. doi: 10.3390/molecules25071487   Rational design, chemical synthesis and biological evaluation of novel biguanides exploring species-specificity responsiveness of TAAR1 agonists. Guariento S, Tonelli M, Espinoza S, Gerasimov AS, Gainetdinov RR, Cichero E.Eur J Med Chem. 2018 Feb 25;146:171-184. doi: 10.1016/j.ejmech.2018.01.059.   Molecular docking studies- please revise this section being more informative. docking results are poorly discussed. The Authors should better detail the experimental protocol they applied, how they identified the binding site, what kind of reference inhibitors discussed in the literature or reported as X ray data They used to validate their docking procedure and results.

Author Response

The manuscript "Design, synthesis, and molecular docking study of new tyrosyl- 2 DNA phosphodiesterase 1 (TDP1) inhibitors combining resin acids and adamantane moieties" seems to be interesting even if it is sometimes poorly discussed and detailed. In my opinion, revisions are necessary prior to publication.

Thank you very much for valuable comments.

Abstract- please modify it in order to be more informative. Revise the last sentence including the main information they achieved through their dockign studies.

The last sentence in Abstract was modified to be more informative: Based on molecular docking results, we suppose that adamantane derivatives of resin acids bind to the TDP1 covalent intermediate, forming a hydrogen bond with Ser463 and hydrophobic contacts with the Phe259 and Trp590 residues and the oligonucleotide fragment of the substrate.

Introduction- please the sentences "Recently, compounds that act as DNA 
repair inhibitors have been considered as potential drugs[1,2]. The enzyme tyrosil-DNA- 36 phosphodiesterase 1 (TDP1) is one of the promising ones[3]". should be accompanied by a scheme about the main chemical scaffold so far developed in this context.

All the following sentence "Hybrid molecules created from different pharmacophores of natural and synthetic 48 equivalents are successfully used in pharmaceutical practice[7]. New hybrid compounds 49
have been synthesised by combining the pharmacophoric moiety of a set of natural com- 50
pounds with inhibitory properties against TDP1. These include phenolic usnic acid and a set of different monoterpenoid fragments[8], 7-hydroxycoumarin and monoterpenoid moieties[9], and 4-arylcoumarin and monoterpenoid fragments[10]. Our group previously obtained a set of ureas and thioureas based on the natural terpenoid dehydroabietylamine[11]". should be supported by the proper chemical series. Please, the related figures.

We thank the reviewer for this comment. As recommended, we have added 2 figures in the introduction clearly demonstrating the structures of known inhibitors. This really makes it clearer which substances are being referred to.

Figure 1- it seems to be unclear as well as the rationale of this study.

Please, the introduction should be concluded with the aims of this study.

 We have added a more detailed description of the purpose of the study in the introduction.

Results and discussion- 2.2. TDP1 assay and cytotoxicity studies

Please, remove the PK and toxicity prediction by ACD/Percepta and move it in a specific paragraph. Please discuss more in details this kind of in silico study. Refer to the following papers:

New Insights into the Binding Features of F508del CFTR Potentiators: A Molecular Docking, Pharmacophore Mapping and QSAR Analysis Approach.

Righetti G, Casale M, Tonelli M, Liessi N, Fossa P, Pedemonte N, Millo E, Cichero E.Pharmaceuticals (Basel). 2020 Dec 4;13(12):445. doi: 10.3390/ph13120445   Synthesis and Biological Evaluation of Novel (thio)semicarbazone-Based Benzimidazoles as Antiviral Agents against Human Respiratory Viruses. Francesconi V, Cichero E, Schenone S, Naesens L, Tonelli M.Molecules. 2020 Mar 25;25(7):1487. doi: 10.3390/molecules25071487   Rational design, chemical synthesis and biological evaluation of novel biguanides exploring species-specificity responsiveness of TAAR1 agonists. Guariento S, Tonelli M, Espinoza S, Gerasimov AS, Gainetdinov RR, Cichero E.Eur J Med Chem. 2018 Feb 25;146:171-184. doi: 10.1016/j.ejmech.2018.01.059. 

Revised as suggested. Please see specific paragraph in “Results and Discussion»: QSAR prediction methods offer a useful tool to identify drug-like compounds [REFs], and therefore we have calculated LogP values for synthesized inhibitors as main determinant of brain tissue binding. Octanol/water LogP predicted with GALAS algorithm [REF] and QSAR software ACD/Percepta (www.acdlabs.com) indicate that the obtained adamantane derivatives have similar lipophilicity (Table 1). The corresponding rate of brain penetration LogPS and extent of brain penetration LogBB, calculated using LogP, molecular size, and H-bonding parameters as inputs, are suitable for penetration into the central nervous system (see Table S1)..

 Molecular docking studies- please revise this section being more informative. docking results are poorly discussed. The Authors should better detail the experimental protocol they applied, how they identified the binding site, what kind of reference inhibitors discussed in the literature or reported as X ray data They used to validate their docking procedure and results.

Corresponding sentences were added in “Materials and Methods”: In our previous study, the TDP1 substrate-binding groove was identified and mapped based on the 1NOP structure (covalent complex with substrate analogue), and successfully tested in docking runs with diazaadamantane derivatives as reference TDP1 inhibitors [REF]. Docking was done using a genetic algorithm in “extra precision” mode. The protein structure was rigid, whereas rotating functional/rotatable groups of ligands was allowed.

Round 2

Reviewer 1 Report

The authors significantly improved their manuscript. Now the article may be published in the Pharmaceuticals journal. As a reviewer, I have no further comments.